# DataBright: Towards a Global Exchange for Decentralized Data Ownership and Trusted Computation

**David Dao[†], Dan Alistarh[‡], Claudiu Musat[∗], Ce Zhang[†]**
[†]ETH Zurich
{david.dao,ce.zhang}@inf.ethz.ch
[‡]IST Austria
dan.alistarh@ist.ac.at
[∗]Swisscom
claudiu.musat@swisscom.com

## Abstract

It is safe to assume that, for the foreseeable future, machine learning, especially deep learning, will remain both *data-* and *computation-hungry*. In this paper, we ask: *Can we build a global exchange where everyone can contribute computation and data to train the next generation of machine learning applications?*

We present an early, but running prototype of DataBright, a system that turns the creation of training examples and the sharing of computation into an investment mechanism. Unlike most crowdsourcing platforms, where the contributor gets paid when they submit their data, DataBright pays dividends *whenever a contributor's data or hardware is used by someone to train a machine learning model*. The contributor becomes a shareholder in the dataset they created. To enable the measurement of usage, a computation platform that contributors can trust is also necessary. DataBright thus merges both a data market and a trusted computation market.

We illustrate that trusted computation can enable the creation of an AI market, where each data point has an exact value that should be paid to its creator. DataBright allows data creators to retain ownership of their contribution and attaches to it a measurable value. The value of the data is given by its utility in subsequent distributed computation done on the DataBright computation market. The computation market allocates tasks and subsequent payments to pooled hardware. This leads to the creation of a decentralized AI cloud. Our experiments show that trusted hardware such as Intel SGX can be added to the usual ML pipeline with no additional costs. We use this setting to orchestrate distributed computation that enables the creation of a computation market. DataBright is available for download at https://github.com/ds3lab/databright.

## 1 Introduction and Motivation

Training modern deep learning models require a vast amount of data and computation, neither of which is cheap to get. Although the availability of crowdsourcing platforms such as Mechanical Turk and cloud platforms such as AWS, in principle, provides a baseline solution to harvest labeled data and computation, there

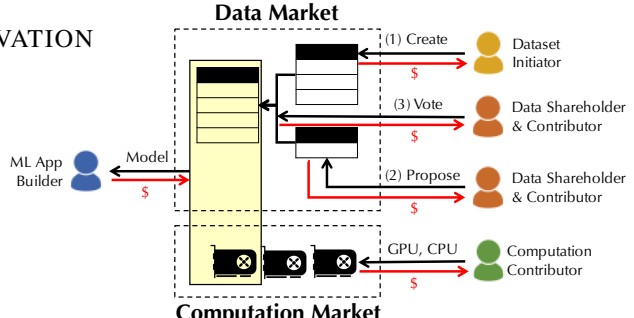

are several limitations. On the data side, the currently access to data is opaque and there is no incentive to share any, while workers are not incentivized to produce high-quality data product. On the computation side, the price of computation on the cloud is still high, while on the other hand,

vast amount of computation resources are being "wasted" for Bitcoin mining. In this work, we are motivated by these limitations, and proposed DATABRIGHT, a decentralized market for data and computation, illustrated in the figure on the right.

## 2    DECENTRALIZED OWNERSHIP OF DATA

In traditional crowdsourcing, workers get a (small) payment for their work and their involvement in the task is over. The people who create the data have no stake in the final result and thus little incentive to do a good job. In DATABRIGHT, we take a different view of data ownership. The ownership of data always initially belongs to the worker who creates the data, who is then free to transfer it. Whenever it gets used, the creators get paid (i.e. the data pay a dividend). We be-

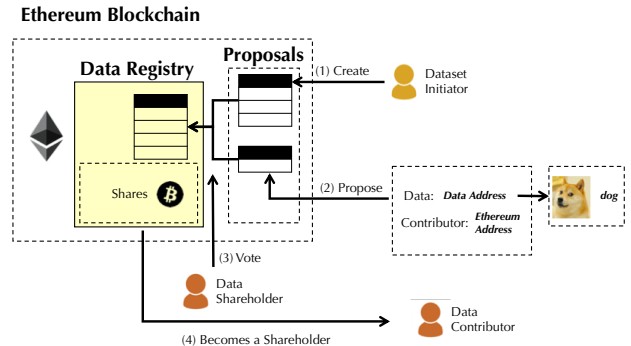

lieve this mechanism can incentivize workers to prioritize the quality and utility of the data over the speed of its creation. In this section, we describe the data market implemented as decentralized application (DApp) using smart contracts over Ethereum.

We provide a data market that allows everyone to be a data contributor. As a first step, a **data initiator** sets up a data request. **Data contributors** collect data and store them at an address outside the blockchain (either in IPFS or a local database). They then submit a data proposal containing their wallet address and the address to their dataset. **Data curators** holding computational tokens are allowed to see and vote on data proposals. They can also forward their voting rights (i.e. tokens) temporarily to an oracle (e.g. an image classifier) that can vote in their stead. If a data proposal reaches a vote threshold it gets accepted and stored as immutable entry into a data registry located on the chosen blockchain. The computation market will use this registry to access datasets and channel payments to the data contributors. Moreover, a new token is minted and **the data contributor becomes a shareholder** in the data registry.

The DApp allows voting to be performed without the need of a third party and consists of three main contracts:

1. **CuratorToken** keeps a list of all token owners, allowing them to vote on data proposals. Tokens can be minted to turn data contributors into shareholders

2. **DataAssociation** is the main contract of DApp. It manages all proposals and can create new "Database" contracts. It is owned by all shareholders

3. **Database** contracts list all accepted data proposals for future queries via the computation market.

The enduring notion of ownership creates a unique challenge — since data can be easily copied, repackaged and resold, the uncontrolled disclosure of data can easily dilute its value. Our view is that the only way of safeguarding against theft is to never disclose the whole data to unauthorized entities. Therefore, along with the data market, DATABRIGHT also has a trusted computation market to process the data (see Section 3).

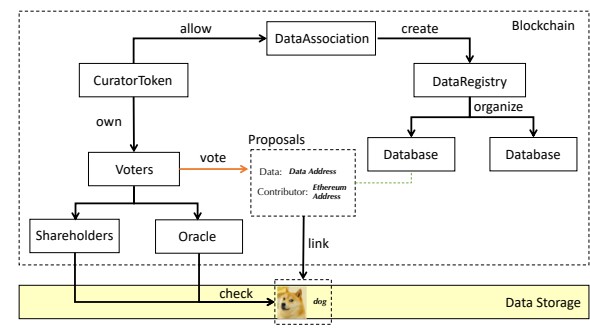

## 3    TRUSTED COMPUTATION MARKET

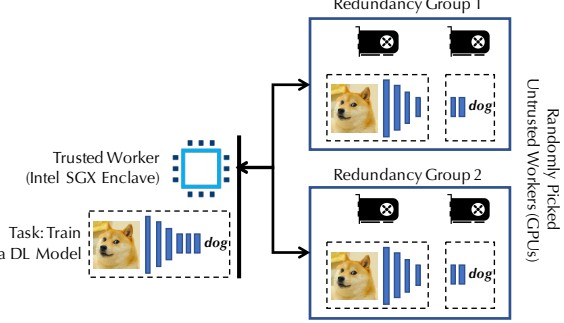

The computation market of DATABRIGHT consists of a pool of computation devices, each of which is owned by a *contributor*. Whenever a contributor's device is used for training a model, the contributor gets paid, either by real money or through a cryptocurrency. (Our testbed is implemented over Ethereum.) The key challenge is providing *trusted computation*: the user, who pays to train her model, needs to have confidence that the model returned by DATABRIGHT is actually the model she asked the system to train. For example, if she is paying $100 to train a ResNet-18 for 100 epochs, she needs a way to prove that DATABRIGHT does not only train 99 epochs, or worse, just returns a list of random numbers.

Recently, Zhang et al. (2017a) have proposed a trusted computation engine built on blockchain with *trusted hardware* (Intel SGX), which can guarantee that each worker faithfully executes their code. However, their design assumes that *all* workers have Intel SGX support enabled. In DATABRIGHT, this assumption is not ideal: for instance, most computation to train a deep learning model happens on NVIDIA GPUs which do not have Intel SGX support.

DATABRIGHT's computation market uses a hybrid model, which assumes a small set of trusted devices, which will be used for scheduling and verification, and a larger pool of untrusted workers, e.g. GPUs, which will be used for the bulk of the work. We implement the following protections to ensure work verification:

1. **Triple modular redundancy (TMR).** Users can choose to have untrusted computation be protected with standard TMR. In a nutshell, trusted devices randomly sample untrusted devices and form three redundancy groups, each of which conducts the same computation. Trusted devices will only return the result to a user when all these redundancy groups return the same result.

2. **Periodical Reallocation.** One possible attack is that an untrusted worker records the data it receives, and resells it. To prevent this, the trusted workers will limit the amount of data an untrusted worker can see to at most 5% of the whole training set.

3. **Model Splitting.** Another concern is that users may wish to avoid a worker having access to the trained model. DATABRIGHT provides an *optional* way to avoid this by splitting the models into pieces, and put each piece on a different randomly sampled untrusted worker. The untrusted workers then communicate the activations and gradients of a single layer; no worker has access to the full model.

## 3.1 EXPERIMENTS

We illustrate the performance overhead introduced by our trusted computation

|  | SGX overhead | Forward Pass | Comm. Time | Epoch Time |
|---|---|---|---|---|
| Standard Training, 1 GPU | N/A | 92 ms/batch | 0 ms/batch | 193 min |
| Standard Training, 2 GPUs | N/A | 52 ms/batch | 0 ms/batch | 118 min |
| No splitting, 1 GPU |  | 92 ms/batch | 0 ms/batch | 193 min |
| No splitting, 2 GPUs | 176.66 ms/run | 46 ms/batch | 2622 ms/batch | 1749 min |
| 2-Way Splitting, 2 GPUs |  | 93 ms/batch | 36 ms/batch | 225 min |

design. We trained a VGG-16 network on ImageNet with three workers – one trusted worker with Intel SGX support, and two untrusted GPU machines. Machines are connected by a (slow) 1Gbps network. When performing *model splitting*, we split into two pieces, at the `fc6` layer and use batch size 32. As shown in the table, executing the scheduling and transmission via Intel SGX/TLS introduces almost negligible overhead. Parallelizing across two nodes (with no splitting) introduces significant communication overhead, due to slow network connection. There is a vast literature exists on reducing these costs Seide et al. (2014); Alistarh et al. (2016); Zhang et al. (2017b).

Model splitting introduces extra overheads compared with using just one GPUs. Part of it is not fundamental and can be fixed by better system optimization — currently, when we split the model in two ways, both GPUs are not always busy, and thus, the computation time can be $2\times$ larger than considering 2 GPUs on a single machine. We could use asynchronous training Lian et al. (2015) to bring this overhead down. Another overhead is the triple modular redundancy. Using this mode in DATABRIGHT introduces roughly $3\times$ overhead to provide trusted compute over untrusted workers; when the user chooses to split the model for better privacy, the current version of DATABRIGHT introduces roughly $6\times$ overhead, although we believe this can be significantly reduced through careful optimization.

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
