# OpenReview forum: "DataBright: Towards a Global Exchange for Decentralized Data Ownership and Trusted Computation"
_ICLR.cc/2018/Workshop — Reject_

### Official Review · AnonReviewer3 · 2018-03-09
**practical usefulness of the system is unconvincing**

**Rating:** 3
**Confidence:** 3

**Review:**

How would users be able to judge the value of data and pay for it before accessing them? Unless this question can be answered, I am not sure how we could convince people to participate in this marketplace. Note that the value of data depends on what data a user already has. For example, ImageNet is a great labeled dataset for machine learning, but for someone who already has the dataset, another copy of ImageNet dataset has zero value. Therefore, if hundreds of people are serving a copy of ImageNet dataset, none of these would be useful for most people, although each dataset by itself is highly valuable.

The guarding mechanism to prevent untrusted workers from stealing training data seems to be too weak to be practical. Authors propose to restrict each untrusted worker from seeing more than 5% of training data, but what if multiple untrusted workers are owned by the same organization?

---

### Official Review · AnonReviewer1 · 2018-03-09
**novel computation offering for ML users**

**Rating:** 6
**Confidence:** 5

**Review:**

This paper introduces data-bright that allows data and computation stakeholders to benefit from their contributions. This is different from the traditional ML cloud offerings where one buys computation from a centralized large cloud provider and arranges for their own data.

Overall, the idea is interesting to generate a discussion amongst the audience. The use of SGX and trusted computation principles is clever. The data contributors are already rewarded in the current research ecosystems via citations that may be worth comparing with the paper’s proposed reward model.

Even though the idea is fairly novel, I find it a very less relevant to the ICLR research community. There is interest here in terms of economics, incentives and systems implementation.  However, the ML community can be a user of this technology and may find this interesting and discussion worthy.

---

### Official Review · AnonReviewer2 · 2018-03-10
**I like what the authors are doing, and hope that they try to implement something like their system**

**Rating:** 6
**Confidence:** 3

**Review:**


Access to data and computational resources are big problems in doing state-of-the-art machine learning research.

The authors have focused very much on creating a data resource.

However, the motivation for this choice seems to have led the authors astray.

The authors' motivation is revealed in this comment: "In traditional crowdsourcing, workers get a (small) payment for their work and  their  involvement  in  the  task  is over.  The people who create the data have no stake in the final result and thus little incentive to do a good job."

But crowd workers are incentived to do good work because they don't want their work to be rejected.  It is common wisdom within the HCI / human computation community that workers will produce good work if you design the task clearly. The fault in getting bad data is almost always on the side of the researcher for not designing a good task. (You can get a sense for this if you watch your requester ratings/reviews on Turkopticon or similar platforms)

The authors motivate their system for computational resources as a way to prevent copying data.

In my opinion, computational resources is actually a much bigger factor than access to data in the type of work that a researcher can do in machine learning. We are surrounded by data, and storage is cheap. But computation remains prohibitively expensive.

For this reason, thinking about how to create a commons for computational resources could be a great idea. The authors don't deal with the economics of their platform, though. Is there a way for it to be a shared resource that can allow under-resourced labs to increase their computational capacity?

That said, there are huge problems with platforms like MTurk in terms of workers rights, and the authors' proposal could help that situation.  Providing ownership of data to crowd workers is a nice idea not just for incentivizing them, but also for empowering them.

It's not clear from the paper how the data infrastructure outside etherium would be supported or maintained. It seems there must be some additional scaffolding software. Who owns and maintains that?

The researchers should also be aware that governing commons is notoriously difficult. Ostrom is a good reference for thinking through the issues at play.

In brief, I like that the authors are thinking creatively about important problems, and it would be great to try to implement something like what the authors describe, but I think there will be many challenges to deploying the particular system the authors have outlined.

---

### Decision · Program_Chairs · 2018-03-20
**ICLR 2018 Workshop Acceptance Decision**

**Decision:**

Reject

**Comment:**

Based on the reviews, this paper has not been accepted for presentation at the ICLR workshop. However, the conversation and updates can continue to appear here on OpenReview.